# Don't Decay the Learning Rate, Increase the Batch Size

**Samuel L. Smith**[*]**, Pieter-Jan Kindermans**[*]**, Chris Ying & Quoc V. Le**
Google Brain
{`slsmith, pikinder, chrisying, qvl`}@google.com

## Abstract

It is common practice to decay the learning rate. Here we show one can usually obtain the same learning curve on both training and test sets by instead increasing the batch size during training. This procedure is successful for stochastic gradient descent (SGD), SGD with momentum, Nesterov momentum, and Adam. It reaches equivalent test accuracies after the same number of training epochs, but with fewer parameter updates, leading to greater parallelism and shorter training times. We can further reduce the number of parameter updates by increasing the learning rate $\epsilon$ and scaling the batch size $B \propto \epsilon$. Finally, one can increase the momentum coefficient $m$ and scale $B \propto 1/(1 - m)$, although this tends to slightly reduce the test accuracy. Crucially, our techniques allow us to repurpose existing training schedules for large batch training with no hyper-parameter tuning. We train ResNet-50 on ImageNet to 76.1% validation accuracy in under 30 minutes.

## 1 Introduction

Stochastic gradient descent (SGD) remains the dominant optimization algorithm of deep learning. However while SGD finds minima that generalize well (Zhang et al., 2016; Wilson et al., 2017), each parameter update only takes a small step towards the objective. Increasing interest has focused on large batch training (Goyal et al., 2017; Hoffer et al., 2017; You et al., 2017a), in an attempt to increase the step size and reduce the number of parameter updates required to train a model. Large batches can be parallelized across many machines, reducing training time. Unfortunately, when we increase the batch size the test set accuracy often falls (Keskar et al., 2016; Goyal et al., 2017).

To understand this surprising observation, Smith & Le (2017) argued one should interpret SGD as integrating a stochastic differential equation. They showed that the scale of random fluctuations in the SGD dynamics, $g = \epsilon(\frac{N}{B} - 1)$, where $\epsilon$ is the learning rate, $N$ training set size and $B$ batch size. Furthermore, they found that there is an optimum fluctuation scale $g$ which maximizes the test set accuracy (at constant learning rate), and this introduces an optimal batch size proportional to the learning rate when $B \ll N$. Goyal et al. (2017) already observed this scaling rule empirically and exploited it to train ResNet-50 to 76.3% ImageNet validation accuracy in one hour. Here we show,

- When one decays the learning rate, one simultaneously decays the scale of random fluctuations $g$ in the SGD dynamics. **Decaying the learning rate is simulated annealing**. We propose an alternative procedure; instead of decaying the learning rate, we increase the batch size during training. This strategy achieves near-identical model performance on the test set with the same number of training epochs but significantly fewer parameter updates. Our proposal does not require any fine-tuning as we follow pre-existing training schedules; when the learning rate drops by a factor of $\alpha$, we instead increase the batch size by $\alpha$.

- As shown previously, we can further reduce the number of parameter updates by increasing the learning rate and scaling $B \propto \epsilon$. One can also increase the momentum coefficient and scale $B \propto 1/(1 - m)$, although this slightly reduces the test accuracy. We train Inception-ResNet-V2 on ImageNet in under 2500 parameter updates, using batches of 65536 images, and reach a validation set accuracy of 77%. We also replicate the setup of Goyal et al. (2017) on TPU and train ResNet-50 on ImageNet to 76.1% accuracy in under 30 minutes.

---

[*]Both authors contributed equally. Work performed as members of the Google Brain Residency Program.

We note that a number of recent works have discussed increasing the batch size during training (Friedlander & Schmidt, 2012; Byrd et al., 2012; Balles et al., 2016; Bottou et al., 2016; De et al., 2017), but to our knowledge no paper has shown empirically that increasing the batch size and decaying the learning rate are quantitatively equivalent. A key contribution of our work is to demonstrate that decaying learning rate schedules can be directly converted into increasing batch size schedules, and vice versa; providing a straightforward pathway towards large batch training.

In section 2 we discuss the convergence criteria for SGD in strongly convex minima, in section 3 we interpret decaying learning rates as simulated annealing, and in section 4 we discuss the difficulties of training with large momentum coefficients. Finally in section 5 we present conclusive experimental evidence that the empirical benefits of decaying learning rates in deep learning can be obtained by instead increasing the batch size during training. We exploit this observation and other tricks to achieve efficient large batch training on CIFAR-10 and ImageNet.

## 2 STOCHASTIC GRADIENT DESCENT AND CONVEX OPTIMIZATION

SGD is a computationally-efficient alternative to full-batch training, but it introduces noise into the gradient, which can obstruct optimization. It is often stated that to reach the minimum of a strongly convex function we should decay the learning rate, such that (Robbins & Monro, 1951):

$$\sum_{i=1}^{\infty} \epsilon_i = \infty, \tag{1}$$

$$\sum_{i=1}^{\infty} \epsilon_i^2 < \infty. \tag{2}$$

$\epsilon_i$ denotes the learning rate at the $i^{th}$ gradient update. Intuitively, equation 1 ensures we can reach the minimum, no matter how far away our parameters are initialized, while equation 2 ensures that the learning rate decays sufficiently quickly that we converge to the minimum, rather than bouncing around it due to gradient noise (Welling & Teh, 2011). However, although these equations appear to imply that the learning rate must decay during training, equation 2 holds only if the batch size is constant.[1] To consider how to proceed when the batch size can vary, we follow recent work by Smith & Le (2017) and interpret SGD as integrating the stochastic differential equation below,

$$\frac{d\omega}{dt} = -\frac{dC}{d\omega} + \eta(t) \tag{3}$$

$C$ represents the cost function (summed over all training examples), and $\omega$ represents the parameters, which evolve in continuous "time" $t$ towards their final values. Meanwhile $\eta(t)$ represents Gaussian random noise, which models the consequences of estimating the gradient on a mini-batch. They showed that the mean $\langle \eta(t) \rangle = 0$ and variance $\langle \eta(t)\eta(t') \rangle = gF(\omega)\delta(t - t')$, where $F(\omega)$ describes the covariances in gradient fluctuations between different parameters. They also proved that the "noise scale" $g = \epsilon(\frac{N}{B} - 1)$, where $\epsilon$ is the learning rate, $N$ the training set size and $B$ the batch size. This noise scale controls the magnitude of the random fluctuations in the training dynamics.

Usually $B \ll N$, and so we may approximate $g \approx \epsilon N/B$. When we decay the learning rate, the noise scale falls, enabling us to converge to the minimum of the cost function (this is the origin of equation 2 above). However we can achieve the same reduction in noise scale *at constant learning rate* by increasing the batch size. The main contribution of this work is to show that it is possible to make efficient use of vast training batches, if one increases the batch size during training at constant learning rate until $B \sim N/10$. After this point, we revert to the use of decaying learning rates.

## 3 SIMULATED ANNEALING AND THE GENERALIZATION GAP

To the surprise of many researchers, it is now increasingly accepted that small batch training often generalizes better to the test set than large batch training. This "generalization gap" was explored extensively by Keskar et al. (2016). Smith & Le (2017) observed an optimal batch size $B_{opt}$ which maximized the test set accuracy at constant learning rate. They argued that this optimal batch size

---

[1]Strictly speaking, equation 2 holds if the batch size is bounded by a value below the training set size.

arises when the noise scale $g \approx \epsilon N/B$ is also optimal, and supported this claim by demonstrating empirically that $B_{opt} \propto \epsilon N$. Earlier, Goyal et al. (2017) exploited a linear scaling rule between batch size and learning rate to train ResNet-50 on ImageNet in one hour with batches of 8192 images.

These results indicate that gradient noise can be beneficial, especially in non-convex optimization. It has been proposed that noise helps SGD escape "sharp minima" which generalize poorly (Hochreiter & Schmidhuber, 1997; Chaudhari et al., 2016; Keskar et al., 2016; Smith & Le, 2017). Given these results, it is unclear to the present authors whether equations 1 and 2 are relevant in deep learning. Supporting this view, we note that most researchers employ early stopping (Prechelt, 1998), whereby we intentionally prevent the network from reaching a minimum. Nonetheless, decaying learning rates are empirically successful. To understand this, we note that introducing random fluctuations whose scale falls during training is also a well established technique in non-convex optimization; simulated annealing. The initial noisy optimization phase allows us to explore a larger fraction of the parameter space without becoming trapped in local minima. Once we have located a promising region of parameter space, we reduce the noise to fine-tune the parameters.

Finally, we note that this interpretation may explain why conventional learning rate decay schedules like square roots or exponential decay have become less popular in deep learning in recent years. Increasingly, researchers favor sharper decay schedules like cosine decay (Loshchilov & Hutter, 2016) or step-function drops (Zagoruyko & Komodakis, 2016). To interpret this shift, we note that it is well known in the physical sciences that slowly annealing the temperature (noise scale) helps the system to converge to the global minimum, which may be sharp. Meanwhile annealing the temperature in a series of discrete steps can trap the system in a "robust" minimum whose cost may be higher but whose curvature is lower. We suspect a similar intuition may hold in deep learning.

## 4   THE EFFECTIVE LEARNING RATE AND THE ACCUMULATION VARIABLE

Many researchers no longer use vanilla SGD, instead preferring SGD with momentum. Smith & Le (2017) extended their analysis of SGD to include momentum, and found that the "noise scale",

$$g \quad = \quad \frac{\epsilon}{1-m}\left(\frac{N}{B}-1\right) \tag{4}$$

$$\approx \quad \frac{\epsilon N}{B(1-m)} \tag{5}$$

This reduces to the noise scale of vanilla SGD when the momentum coefficient $m \to 0$. Intuitively, $\epsilon_{eff} = \epsilon/(1-m)$ is the effective learning rate. They proposed to reduce the number of parameter updates required to train a model by increasing the learning rate and momentum coefficient, while simultaneously scaling $B \propto \epsilon/(1-m)$. We find that increasing the learning rate and scaling $B \propto \epsilon$ performs well. However increasing the momentum coefficient while scaling $B \propto 1/(1-m)$ slightly reduces the test accuracy. To analyze this observation, consider the momentum update equations,

$$\Delta A \quad = \quad -(1-m)A + \frac{d\hat{C}}{d\omega}, \tag{6}$$

$$\Delta \omega \quad = \quad -A\epsilon. \tag{7}$$

$A$ is the "accumulation", while $\frac{d\hat{C}}{d\omega}$ is the mean gradient per training example, estimated on a batch of size $B$. In Appendix A we analyze the growth of the accumulation at the start of training. This variable tracks the exponentially decaying average of gradient estimates, but initially it is initialized to zero. We find that the accumulation grows in exponentially towards its steady state value over a "timescale" of approximately $B/(N(1-m))$ training epochs. During this time, the magnitude of the parameter updates $\Delta \omega$ is suppressed, reducing the rate of convergence. Consequently when training at high momentum one must introduce additional epochs to allow the dynamics to catch up.

Furthermore, when we increase the momentum coefficient we increase the timescale required for the accumulation to forget old gradients (this timescale is also $\sim B/(N(1-m))$). Once this timescale becomes several epochs long, the accumulation cannot adapt to changes in the loss landscape, impeding training. This is likely to be particularly problematic at points where the noise scale decays. Kingma & Ba (2014) proposed initialization bias correction, whereby the learning rate is increased at early times to compensate the suppressed initial value of the accumulation. However when the batch size is large, we found that this often causes instabilities during the early stages of training. We note that Goyal et al. (2017) recommended a reduced learning rate for the first few epochs.

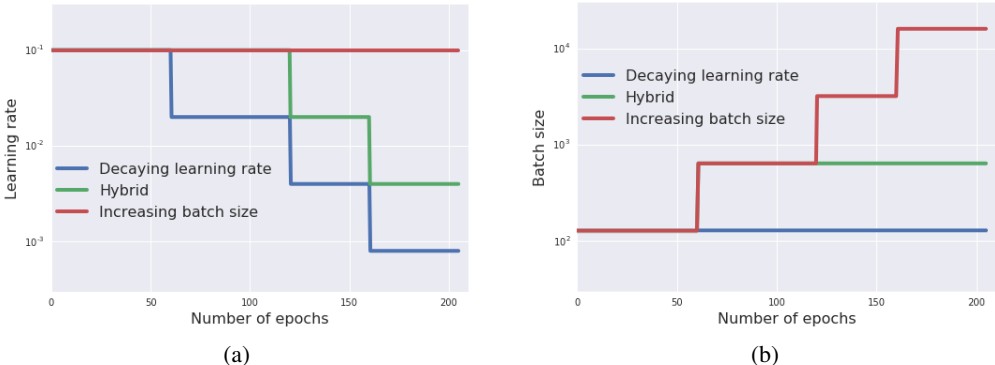

Figure 1: Schedules for the learning rate (a) and batch size (b), as a function of training epochs.

## 5 EXPERIMENTS

In section 5.1, we demonstrate that decreasing the learning rate and increasing the batch size during training are equivalent. In section 5.2, we show we can further reduce the number of parameter updates by increasing the effective learning rate and scaling the batch size. In section 5.3 we apply our insights to train Inception-ResNet-V2 on ImageNet, using vast batches of up to 65536 images. Finally in section 5.4, we train ResNet-50 to 76.1% ImageNet validation accuracy within 30 minutes.

### 5.1 SIMULATED ANNEALING IN A WIDE RESNET

Our first experiments are performed on CIFAR-10, using a "16-4" wide ResNet architecture, following the implementation of Zagoruyko & Komodakis (2016). We use ghost batch norm (Hoffer et al., 2017), with a ghost batch size of 128. This ensures the mean gradient is independent of batch size, as required by the analysis of Smith & Le (2017). To demonstrate the equivalence between decreasing the learning rate and increasing the batch size, we consider three different training schedules, as shown in figure 1. "Decaying learning rate" follows the original implementation; the batch size is constant, while the learning rate repeatedly decays by a factor of 5 at a sequence of "steps". "Hybrid" holds the learning rate constant at the first step, instead increasing the batch size by a factor of 5. However after this first step, the batch size is constant and the learning rate decays by a factor of 5 at each subsequent step. This schedule mimics how one might proceed if hardware imposes a limit on the maximum achievable batch size. In "Increasing batch size", we hold the learning rate constant throughout training, and increase the batch size by a factor of 5 at every step.

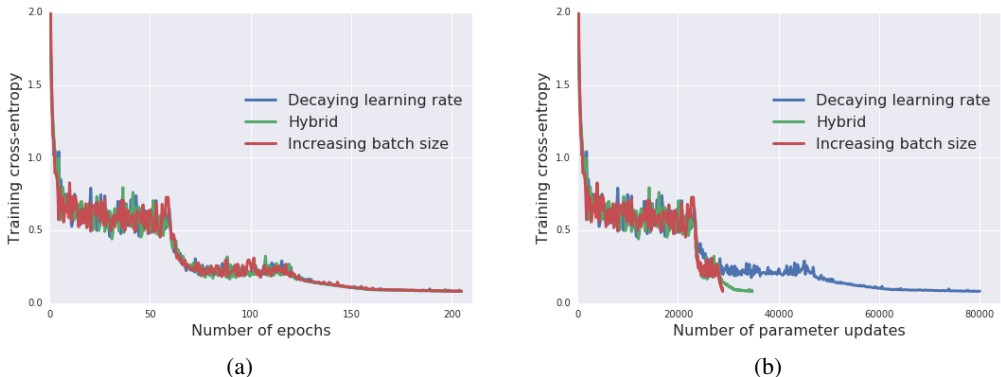

Figure 2: Wide ResNet on CIFAR10. Training set cross-entropy, evaluated as a function of the number of training epochs (a), or the number of parameter updates (b). The three learning curves are identical, but increasing the batch size reduces the number of parameter updates required.

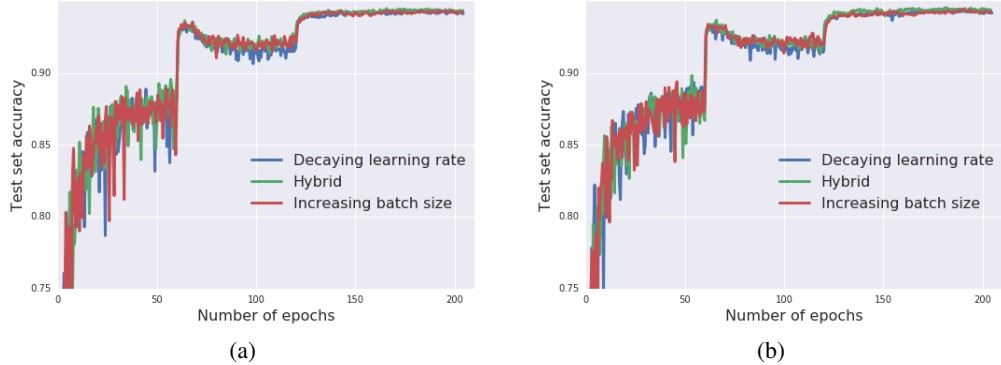

(a)             (b)

Figure 3: Wide ResNet on CIFAR10. Test accuracy during training, for SGD with momentum (a), and Nesterov momentum (b). In both cases, all three schedules track each other extremely closely.

If the learning rate itself must decay during training, then these schedules should show different learning curves (as a function of the number of training epochs) and reach different final test set accuracies. Meanwhile if it is the noise scale which should decay, all three schedules should be indistinguishable. We plot the evolution of the training set cross entropy in figure 2a, where we train using SGD with momentum and a momentum parameter of 0.9. The three training curves are almost identical, despite showing marked drops as we pass through the first two steps (where the noise scale is reduced). These results suggest that it is the noise scale which is relevant, not the learning rate.

To emphasize the potential benefits of increasing the batch size, we replot the training cross-entropy in figure 2b, but as a function of the number of parameter updates rather than the number of epochs. While all three schedules match up to the first "step", after this point increasing the batch size dramatically reduces the number of parameter updates required to train the model. Finally, to confirm that our alternative learning schedules generalize equally well to the test set, in figure 3a we exhibit the test set accuracy, as a function of the number of epochs (so each curve can be directly compared). Once again, the three schedules are almost identical. We conclude that we can achieve all of the benefits of decaying the learning rate in these experiments by instead increasing the batch size.

We present additional results to establish that our proposal holds for a range of optimizers, all using the schedules presented in figure 1. In figure 3b, we present the test set accuracy, when training with Nesterov momentum (Nesterov, 1983) and momentum parameter 0.9, observing three near-identical curves. In figure 4a, we repeat the same experiment with vanilla SGD, again obtaining three highly similar curves (In this case, there is no clear benefit of decaying the learning rate after the first step). Finally in figure 4b we repeat the experiment with Adam (Kingma & Ba, 2014). We

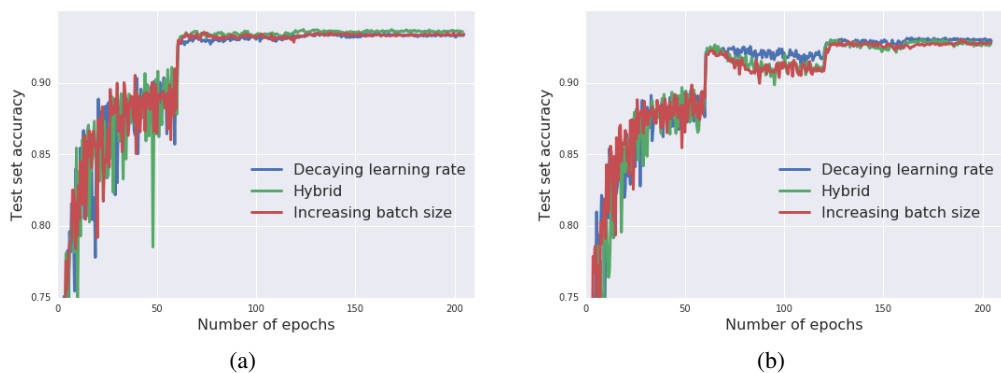

(a)             (b)

Figure 4: Wide ResNet on CIFAR10. The test set accuracy during training, for vanilla SGD (a) and Adam (b). Once again, all three schedules result in equivalent test set performance.

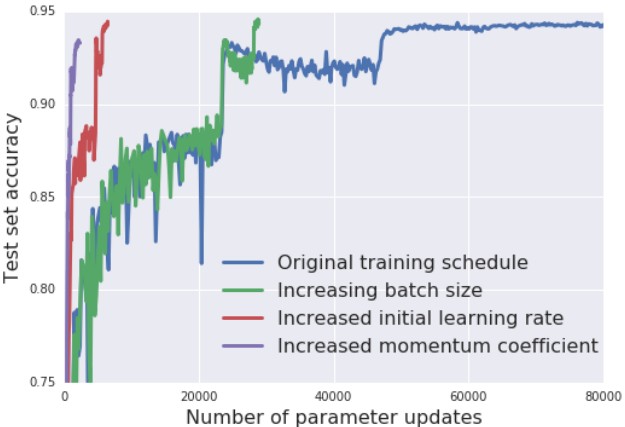

Figure 5: Wide ResNet on CIFAR10. Test accuracy as a function of the number of parameter updates. "Increasing batch size" replaces learning rate decay by batch size increases. "Increased initial learning rate" additionally increases the initial learning rate from 0.1 to 0.5. Finally "Increased momentum coefficient" also increases the momentum coefficient from 0.9 to 0.98.

use the default parameter settings of TensorFlow, such that the initial base learning rate here was $10^{-3}$, $\beta_1 = 0.9$ and $\beta_2 = 0.999$. Thus the learning rate schedule is obtained by dividing figure 1a by $10^{-2}$. Remarkably, even here the three curves closely track each other.

## 5.2 INCREASING THE EFFECTIVE LEARNING RATE

We now focus on our secondary objective; minimizing the number of parameter updates required to train a model. As shown above, the first step is to replace decaying learning rates by increasing batch sizes. We show here that we can also increase the effective learning rate $\epsilon_{eff} = \epsilon/(1 - m)$ at the start of training, while scaling the initial batch size $B \propto \epsilon_{eff}$. All experiments are conducted using SGD with momentum. There are 50000 images in the CIFAR-10 training set, and since the scaling rules only hold when $B \ll N$, we decided to set a maximum batch size $B_{max} = 5120$.

We consider four training schedules, all of which decay the noise scale by a factor of five in a series of three steps. "Original training schedule" follows the implementation of Zagoruyko & Komodakis (2016), using an initial learning rate of 0.1 which decays by a factor of 5 at each step, a momentum coefficient of 0.9, and a batch size of 128. "Increasing batch size" also uses a learning rate of 0.1, initial batch size of 128 and momentum coefficient of 0.9, but the batch size increases by a factor of 5 at each step. These schedules are identical to "Decaying learning rate" and "Increasing batch size" in section 5.1 above. "Increased initial learning rate" also uses increasing batch sizes during training, but additionally uses an initial learning rate of 0.5 and an initial batch size of 640. Finally "Increased momentum coefficient" combines increasing batch sizes during training and the increased initial learning rate of 0.5, with an increased momentum coefficient of 0.98, and an initial batch size of 3200. Note that we only increase the batch size until it reaches $B_{max}$, after this point we achieve subsequent decays in noise scale by decreasing the learning rate. We emphasize that, as in the previous section, all four schedules require the same number of training epochs.

We plot the evolution of the test set accuracy in figure 5, as a function of the number of parameter updates. Our implementation of the original training schedule requires ~80000 updates, and reaches a final test accuracy of 94.3% (the original paper reports 95% accuracy, which we have not been able to replicate). "Increasing batch size" requires ~29000 updates, reaching a final accuracy of 94.4%. "Increased initial learning rate" requires under 6500 updates, reaching a final accuracy of 94.5%. Finally, "Increased momentum coefficient" requires less than 2500 parameter updates, but reaches a lower test accuracy of 93.3%. Across five additional training runs for each schedule, the median accuracies were 94.3%, 94.2%, 94.2% and 93.5% respectively. We discussed a potential explanation for the performance drop when training with large momentum coefficients in section 4. We provide additional results in appendix B, varying the initial learning rate between 0.1 and 3.2 while holding the batch size constant. We find that the test accuracy falls for initial learning rates larger than ~0.4.

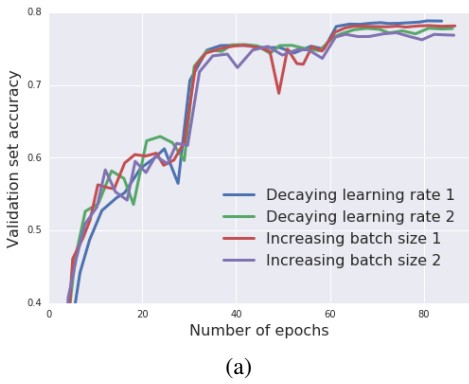 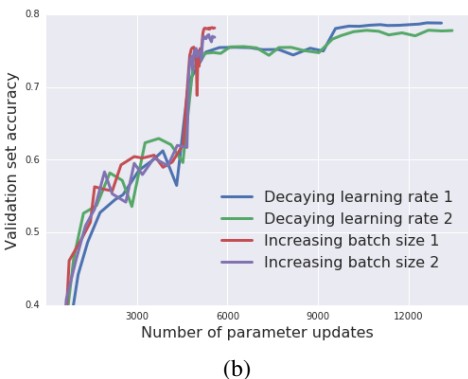

(a)                  (b)

Figure 6: Inception-ResNet-V2 on ImageNet. Increasing the batch size during training achieves similar results to decaying the learning rate, but it reduces the number of parameter updates from just over 14000 to below 6000. We run each experiment twice to illustrate the variance.

## 5.3 TRAINING IMAGENET IN 2500 PARAMETER UPDATES

We now apply our insights to reduce the number of parameter updates required to train ImageNet. Goyal et al. (2017) trained a ResNet-50 on ImageNet in one hour, reaching 76.3% validation accuracy. To achieve this, they used batches of 8192, with an initial learning rate of 3.2 and a momentum coefficient of 0.9. They completed 90 training epochs, decaying the learning rate by a factor of ten at the 30th, 60th and 80th epoch. ImageNet contains around 1.28 million images, so this corresponds to ∼14000 parameter updates. They also introduced a warm-up phase at the start of training, in which the learning rate and batch size was gradually increased.

We also train for 90 epochs and follow the same schedule, decaying the noise scale by a factor of ten at the 30th, 60th and 80th epoch. However we did not include a warm-up phase. To set a stronger baseline, we replaced ResNet-50 by Inception-ResNet-V2 (Szegedy et al., 2017). Initially we used a ghost batch size of 32. In figure 6, we train with a learning rate of 3.0 and a momentum coefficient of 0.9. The initial batch size was 8192. For "Decaying learning rate", we hold the batch size fixed and decay the learning rate, while in "Increasing batch size" we increase the batch size to 81920 at the first step, but decay the learning rate at the following two steps. We repeat each schedule twice, and find that all four runs exhibit a very similar evolution of the test set accuracy during training. The final accuracies of the two "Decaying learning rate" runs are 78.7% and 77.8%, while the final accuracy of the two "Increasing batch size" runs are 78.1% and 76.8%. Although there is a slight drop, the difference in final test accuracies is similar to the variance between training runs. Increasing the batch size reduces the number of parameter updates during training from just over 14000 to below 6000. Note that the training curves appear unusually noisy because we reduced the number of test set evaluations to reduce the model training time.

Goyal et al. (2017) already increased the learning rate close to its maximum stable value. To further reduce the number of parameter updates we must increase the momentum coefficient. We introduce a maximum batch size, $B_{max} = 2^{16} = 65536$. This ensures $B \ll N$, and it also improved the stability of our distributed training. We also increased the ghost batch size to 64, matching the batch size of our GPUs and reducing the training time. We compare three different schedules, all of which have the same base schedule, decaying the noise scale by a factor of ten at the 30th, 60th and 80th epoch. We use an initial learning rate of 3 throughout. "Momentum 0.9" uses an initial batch size of 8192, "Momentum 0.975" uses an initial batch size of 16384, and "Momentum 0.9875" uses an initial batch size of 32768. For all schedules, we decay the noise scale by increasing the batch size until reaching $B_{max}$, and then decay the learning rate. We plot the test set accuracy in figure 7. "Momentum 0.9" achieves a final accuracy of 78.8% in just under 6000 updates. We performed two runs of "Momentum 0.95", achieving final accuracies of 78.1% and 77.8% in under 3500 updates. Finally "Momentum 0.975" achieves final accuracies of 77.5% and 76.8% in under 2500 updates.

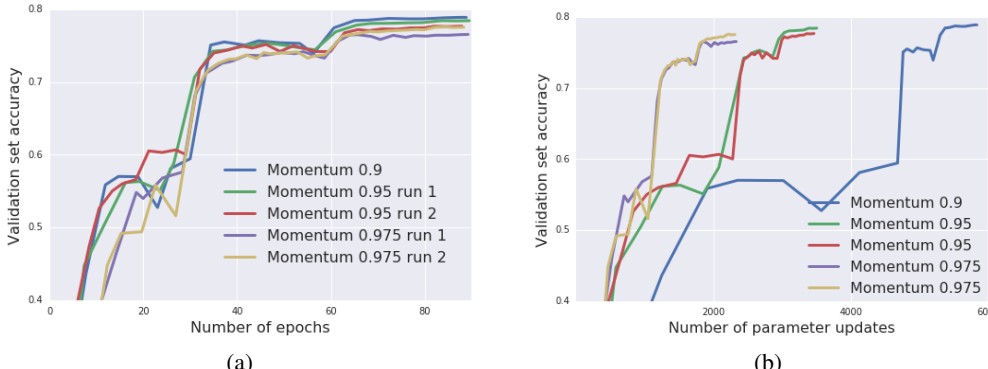

Figure 7: Inception-ResNet-V2 on ImageNet. Increasing the momentum parameter reduces the number of parameter updates required, but it also leads to a small drop in final test accuracy.

### 5.4 TRAINING IMAGENET IN 30 MINUTES

To confirm that increasing the batch size during training can reduce model training times, we replicated the set-up described by Goyal et al. (2017) on a half TPU pod, comprising 256 tensorcores (Jouppi et al., 2017). Using tensorFlow, we first train ResNet-50 for 90 epochs to 76.1% validation set accuracy in under 45 minutes, utilising batches of 8192 images. To utilise the full TPU pod, we then increase the batch size after the first 30 epochs to 16384 images, and achieve the same validation accuracy of 76.1% in under 30 minutes. The last 60 epochs and the first 30 epochs both take just under 15 minutes, demonstrating near-perfect scaling efficiency across the pod, such that the number of parameter updates provides a meaningful measure of the training time. To our knowledge, this is the first procedure which has reduced the training time of Goyal et al. (2017) without sacrificing final validation accuracy (You et al., 2017b; Akiba et al., 2017). By contrast, doubling the initial learning rate and using batches of 16384 images throughout training achieves a lower validation set accuracy of 75.0% in 22 minutes, demonstrating that increasing the batch size during training is crucial to the performance gains above. These results show that the ideas presented in this paper will become increasingly important as new hardware for large-batch training becomes available.

## 6 RELATED WORK

This paper extends the analysis of SGD in Smith & Le (2017) to include decaying learning rates. Mandt et al. (2017) also interpreted SGD as a stochastic differential equation, in order to discuss how SGD could be modified to perform approximate Bayesian posterior sampling. However they state that their analysis holds only in the neighborhood of a minimum, while Keskar et al. (2016) showed that the beneficial effects of noise are most pronounced at the start of training. Li et al. (2017) proposed the use of control theory to set the learning rate and momentum coefficient.

Goyal et al. (2017) observed a linear scaling rule between batch size and learning rate, $B \propto \epsilon$, and used this rule to reduce the time required to train ResNet-50 on ImageNet to one hour. To our knowledge, this scaling rule was fist adopted by Krizhevsky (2014). Bottou et al. (2016) (section 4.2) demonstrated that SGD converges to strongly convex minima in similar numbers of training epochs if $B \propto \epsilon$. Hoffer et al. (2017) proposed an alternative scaling rule, $B \propto \sqrt{\epsilon}$.

You et al. (2017a) proposed Layer-wise Adaptive Rate Scaling (LARS), which applies different learning rates to different parameters in the network, and used it to train ImageNet in 14 minutes (You et al., 2017b), albeit to a lower final accuracy of 74.9%. K-FAC (Martens & Grosse, 2015) is also gaining popularity as an efficient alternative to SGD. Wilson et al. (2017) argued that adaptive optimization methods tend to generalize less well than SGD and SGD with momentum (although they did not include K-FAC in their study), while our work reduces the gap in convergence speed. Asynchronous-SGD is another popular strategy, which enables the use of multiple GPUs even when batch sizes are small (Recht et al., 2011; Dean et al., 2012). We do not consider asynchronous-SGD in this work, since the scaling rules enabled us to use batch sizes on the order of the training set size.

## 7 CONCLUSIONS

We can often achieve the benefits of decaying the learning rate by instead increasing the batch size during training. We support this claim with experiments on CIFAR-10 and ImageNet, and with a range of optimizers including SGD, Momentum and Adam. Our findings enable the efficient use of vast batch sizes, significantly reducing the number of parameter updates required to train a model. This has the potential to dramatically reduce model training times. We further increase the batch size $B$ by increasing the learning rate $\epsilon$ and momentum parameter $m$, while scaling $B \propto \epsilon/(1-m)$. Combining these strategies, we train Inception-ResNet-V2 on ImageNet to 77% validation accuracy in under 2500 parameter updates, using batches of 65536 images. We also exploit increasing batch sizes to train ResNet-50 to 76.1% ImageNet validation set accuracy on TPU in under 30 minutes. Most strikingly, we achieve this without any hyper-parameter tuning, since our scaling rules enable us to directly convert existing hyper-parameter choices from the literature for large batch training.

ACKNOWLEDGMENTS

We thank Prajit Ramachandran, Gabriel Bender, Matthew Johnson and Martin Abadi for helpful discussions. We also thank Vijay Vasudevan, Brennan Saeta, Jonathan Hseu, Bjarke Roune and the rest of the TPU team for technical support.

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

## A    THE GROWTH OF THE ACCUMULATION AT THE START OF TRAINING

The update equations for SGD with momentum,

$$\Delta A \quad = \quad -(1-m)A + \frac{d\hat{C}}{d\omega}, \tag{8}$$

$$\Delta \omega \quad = \quad -A\epsilon. \tag{9}$$

$A$ is the "accumulation" variable, while $\frac{d\hat{C}}{d\omega}$ is the mean gradient per training example, estimated on a batch of size $B$. We initialize the accumulation to zero, and it takes a number of updates for the magnitude of the accumulation to "grow in". During this time, the size of the parameter updates $\Delta\omega$ is suppressed, reducing the effective learning rate. We can model the growth of the accumulation by assuming that the gradient at the start of training is approximately constant, such that $\frac{d\hat{C}}{d\omega} \approx G$. Consequently the accumulation integrates an underlying differential equation,

$$\frac{dA}{ds} = -(1-m)A + G. \tag{10}$$

The variable $s$ describes the number of parameter updates performed. Since $A(0) = 0$, this differential equation has solution, $A = \frac{G}{1-m}\left(1 - e^{-(1-m)s}\right)$. We note that $s = (N/B)N_{epochs}$ to obtain,

$$A = \frac{G}{1-m}\left(1 - e^{-(1-m)(N/B)N_{epochs}}\right). \tag{11}$$

$N_{epochs}$ denotes the number of training epochs performed. The accumulation variable grows in exponentially, and consequently we can estimate the effective number of "lost" training epochs,

$$N_{lost} \quad = \quad \int_0^{\infty} e^{-(1-m)(N/B)N_{epochs}} dN_{epochs} \tag{12}$$

$$= \quad \frac{B}{N(1-m)} \tag{13}$$

Since the batch size $B \propto \epsilon/(1-m)$, we find $N_{lost} \propto \epsilon/(N(1-m)^2)$. We must either introduce additional training epochs to compensate, or ensure that the number of lost training epochs is negligible, when compared to the total number of training epochs performed before the decaying the noise scale. Note that $N_{lost}$ rises most rapidly when one increases the momentum coefficient.

## B    INCREASING THE INITIAL LEARNING RATE

We exhibit the test accuracy of our "16-4" wide ResNet implementation on CIFAR10 in figure 8, as a function of the initial learning rate. For learning rate $\epsilon = 0.1$, the batch size $B = 128$ is constant throughout training. This matches the "Original training schedule" of section 5.2 of the main text. When we increase the learning rate we scale $B \propto \epsilon$ and perform the same number of training epochs.

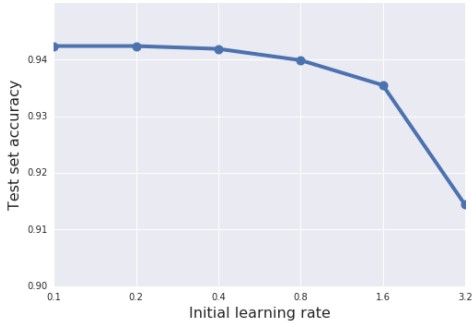

Figure 8: Wide ResNet on CIFAR10. We can only increase the initial learning rate to $\sim 0.4$ before the final test accuracy starts to fall. Each point provided represents the median of five runs.

