# OpenReview forum: "Don't Decay the Learning Rate, Increase the Batch Size"
_ICLR.cc/2018/Conference — Accept (Poster)_

### Official Review · AnonReviewer3 · 2017-11-27
**A simple, relevant observation as computing resources are becoming increasingly available for rent**

**Rating:** 6
**Confidence:** 4

**Review:**

The paper analyzes the the effect of increasing the batch size in stochastic gradient descent as an alternative to reducing the learning rate, while keeping the number of training epochs constant. This has the advantage that the training process can be better parallelized, allowing for faster training if hundreds of GPUs are available for a short time. The theory part of the paper briefly reviews the relationship between learning rate, batch size, momentum coefficient, and the noise scale in stochastic gradient descent. In the experimental part, it is shown that the loss function and test accuracy depend only on the schedule of the decaying noise scale over training time, and are independent of whether this decaying noise schedule is achieved by a decaying learning rate or an increasing batch size. It is shown that simultaneously increasing the momentum parameter and the batch size also allows for fewer parameters, albeit at the price of some loss in performance.

COMMENTS:

The paper presents a simple observation that seems very relevant especially as computing resources are becoming increasingly available for rent on short time scales. The observation is explained well and substantiated by clear experimental evidence. The main issue I have is with the part about momentum. The paragraph below Eq. 7 provides a possible explanation for the performance drop when $m$ is increased. It is stated that at the beginning of the training, or after increasing the batch size, the magnitude of parameter updates is suppressed because $A$ has to accumulate gradient signals over a time scale $B/(N(1-m))$. The conclusion in the paper is that training at high momentum requires additional training epochs before $A$ reaches its equilibrium value. This effect is well known, but it can easily be remedied. For example, the update equations in Adam were specifically designed to correct for this effect. The mechanism is called "bias-corrected moment estimate" in the Adam paper, arXiv:1412.6980. The correction requires only two extra multiplications per model parameter and update step. Couldn't the same or a very similar trick be used to correctly rescale $A$ every time one increases the batch size? It would be great to see the equivalent of Figure 7 with correctly rescaled $A$.

Minor issues:
* The last paragraph of Section 5 refers to a figure 8, which appears to be missing.
* In Eqs. 4 & 5, the momentum parameter $m$ is not yet defined (it will be defined in Eqs. 6 & 7 below).
* It appears that a minus sign is missing in Eq. 7. The update steps describe gradient ascent.
* Figure 3 suggests that most of the time between the first and second change of the noise scale (approx. epochs 60 to 120) are spent on overfitting. This suggests that the number of updates in this segment was chosen unnecessarily large to begin with. It is therefore not surprising that reducing the number of updates does not deteriorate the test set accuracy.
* It would be interesting to see a version of figure 5 where the horizontal axis is the number of epochs. While reducing the number of updates allows for faster training if a large number of parallel hardware instances are available, the total cost of training is still governed by the number of training epochs.
* It appears like the beginning of the second paragraph in Section 5.2 describes figure 1. Is this correct?

---

> ### Author Response · Authors · 2017-12-21
> **Response to review**
>
> We thank the reviewer for their positive review,
>
> We will edit our discussion of momentum in section 4 to explain the problem more clearly. We are currently running experiments to double check, but we do not believe that the “bias-corrected moment estimate” trick will remove the performance gap when training at very large momentum coefficient. This is for two reasons:
>
> 1) When one uses momentum, one introduces a new timescale into the dynamics, the time required for the direction of the parameter updates to change/forget old gradients. When one trains with large batch sizes and large momentum coefficients, this timescale becomes several epochs long. This invalidates the scaling rules, which assume this timescale is negligible. This issue arises throughout training, not just at initialization/after changing the noise scale.
>
> 2) The “bias-corrected moment estimate” ensures that the expected magnitude of the parameter update at the start of training is correct, but it does not ensure that the variance in this parameter update is correct. As a result, bias correction introduces a very large noise scale at the start of training, which decays as the bias correction term falls. The same issue will arise if we used bias correction to reset the accumulation during training at a noise scale step; in fact it would temporarily increase the noise scale every time we try to reduce it.
>
> Responding to the minor issues raised:
> i) Our apologies, this should be figure 7b, we will fix it.
> ii) The momentum coefficient is defined in the first line of the paragraph following eqns 4/5.
> iii) Yes, we will fix this.
> iv) We will check our conclusions hold when we reduce the number of epochs here, however we keep to pre-existing schedules in the paper to emphasize that our techniques can be applied without hyper-parameter tuning.
> v) All curves in figure 5 saw the same number of training epochs.
> vi) The first two schedules described in this paragraph match figure 1, however the following two schedules are new.

---

### Official Review · AnonReviewer1 · 2017-11-27
**Useful empirical validation.**

**Rating:** 7
**Confidence:** 4

**Review:**

The paper represents an empirical validation of the well-known idea (it was published several times before)
to increase the batch size over time. Inspired by recent works on large-batch studies, the paper suggests to adapt the learning rate as a function of the batch size.

I am interested in the following experiment to see how useful it is to increase the batch size compared to fixed batch size settings.

1) The total budget / number of training samples is fixed.
2) Batch size is scheduled to change between B_min and B_max
3) Different setting of B_min and B_max>=B_min are considered, e.g., among [64, 128, 256, 512, ...] or [64, 256, 1024, ...] if it is too expensive.
4) Drops of the learning rates are scheduled to happen at certain times represented in terms of the number of training samples passed so far (not parameter updates).
5) Learning rates and their drops should be rescaled taking into account the schedule of the batch size and the rules to adapt learning rates in large-scale settings as by Goyal.

---

> ### Author Response · Authors · 2017-12-21
> **Response to review**
>
> We thank the reviewer for their positive review.
>
> We'd like to emphasize that our paper verifies a stronger claim than previous works. While previous papers have proposed increasing the batch size over time instead of decaying the learning rate, our work demonstrates that we can directly convert decaying learning rate schedules into increasing batch size schedules and vice-versa; obtaining identical learning curves on both training and test sets for the same number of training epochs seen. To do so, we replace decaying the learning rate by a factor q by increasing the batch size by the same factor q. This strategy allows us to convert between small and large batch training schedules without hyper-parameter tuning, which enabled us to achieve efficient large batch training, with batches of 65,000 examples on ImageNet.
>
> We may have misunderstood, but we believe that we provided the experiment suggested in the review in section 5.1 (figures 1,2 and 3). We consider three schedules, each of which decay the noise scale by a factor of 5 after ~60, ~120 and ~160 epochs. Each schedule sees the same number of training examples. The “decaying learning rate schedule” achieves this by using a constant batch size of 128 and decaying the learning rate by a factor of 5 at each step. The “increasing batch schedule” holds the learning rate fixed and increases the batch size by a factor of 5 at the same steps. Finally the “hybrid” schedule is mix of the two strategies. All three curves achieve identical training curves in terms of number of examples seen (figure 2a), and achieve identical final test accuracy (figure 3a). In this sense, decaying the learning rate and increasing the batch size are identical; they require the same amount of computation to reach the same training/test accuracies. However if one increases the batch size one can benefit from greater parallelism to reduce wall clock time.

---

### Official Review · AnonReviewer2 · 2017-11-27
**reasonable empirical evidence for a not-too-surprising claim / could improve with more diverse set of tasks, wallclock metrics**

**Rating:** 6
**Confidence:** 4

**Review:**

## Review Summary

Overall, the paper's paper core claim, that increasing batch sizes at a linear
rate during training is as effective as decaying learning rates, is
interesting but doesn't seem to be too surprising given other recent work in
this space. The most useful part of the paper is the empirical evidence to
backup this claim, which I can't easily find in previous literature. I wish
the paper had explored a wider variety of dataset tasks and models to better
show how well this claim generalizes, better situated the practical benefits
of the approach (how much wallclock time is actually saved? how well can it be
integrated into a distributed workflow?), and included some comparisons with
other recent recommended ways to increase batch size over time.


## Pros / Strengths

+ effort to assess momentum / Adam / other modern methods

+ effort to compare to previous experimental setups


## Cons / Limitations

- lack of wallclock measurements in experiments

- only ~2 models / datasets examined, so difficult to assess generalization

- lack of discussion about distributed/asynchronous SGD


## Significance

Many recent previous efforts have looked at the importance of batch sizes
during training, so topic is relevant to the community. Smith and Le (2017)
present a differential equation model for the scale of gradients in SGD,
finding a linear scaling rule proportional to eps N/B, where eps = learning
rate, N = training set size, and B = batch size. Goyal et al (2017) show how
to train deep models on ImageNet effectively with large (but fixed) batch
sizes by using a linear scaling rule.

A few recent works have directly tested increasing batch sizes during
training. De et al (AISTATS 2017) have a method for gradually increasing batch
sizes, as do Friedlander and Schmidt (2012). Thus, it is already reasonable to
practitioners that the proposed linear scaling of batch sizes during training
would be effective.

While increasing batch size at the proposed linear scale is simple and seems
to be effective, a careful reader will be curious how much more could be
gained from the backtracking line search method proposed in De et al.


## Quality

Overall, only single training runs from a random initialization are used. It
would be better to take the best of many runs or to somehow show error bars,
to avoid the reader wondering whether gains are due to changes in algorithm or
to poor exploration due to bad initialization. This happens a lot in Sec. 5.2.

Some of the experimental setting seem a bit haphazard and not very systematic.
In Sec. 5.2, only two learning rate scales are tested (0.1 and 0.5). Why not
examine a more thorough range of values?

Why not report actual wallclock times? Of course having reduced number of
parameter updates is useful, but it's difficult to tell how big of a win this
could be.

What about distributed SGD or asyncronous SGD (hogwild)? Small batch sizes
sometimes make it easier for many machines to be working simultaneously. If we
scale up to batch sizes of ~ N/10, we can only get 10x speedups in
parallelization (in terms of number of parameter updates). I think there is
some subtle but important discussion needed on how this framework fits into
modern distributed systems for SGD.


## Clarity

Overall the paper reads reasonably well.

Offering a related work "feature matrix" that helps readers keep track of how
previous efforts scale learning rates or minibatch sizes for specific
experiments could be valueable. Right now, lots of this information is just
provided in text, so it's not easy to make head-to-head comparisons.

Several figure captions should be updated to clarify which model and dataset
are studied. For example, when skimming Fig. 3's caption there is no such
information.

## Paper Summary

The paper examines the influence of batch size on the behavior of stochastic
gradient descent to minimize cost functions. The central thesis is that
instead of the "conventional wisdom" to fix the batch size during training and
decay the learning rate, it is equally effective (in terms of training/test
error reached) to gradually increase batch size during training while fixing
the learning rate. These two strategies are thus "equivalent". Furthermore,
using larger batches means fewer parameter updates per epoch, so training is
potentially much faster.

Section 2 motivates the suggested linear scaling using previous SGD analysis
from Smith and Le (2017). Section 3 makes connections to previous work on
finding optimal batch sizes to close the generaization gap. Section 4 extends
analysis to include SGD methods with momentum.

In Section 5.1, experiments training a 16-4 ResNet on CIFAR-10 compare three
possible SGD schedules: * increasing batch size * decaying learning rate *
hybrid (increasing batch size and decaying learning rate) Fig. 2, 3 and 4 show
that across a range of SGD variants (+/- momentum, etc) these three schedules
have similar error vs. epoch curves. This is the core claimed contribution:
empirical evidence that these strategies are "equivalent".

In Section 5.3, experiments look at Inception-ResNet-V2 on ImageNet, showing
the proposed approach can reach comparable accuracies to previous work at even
fewer parameter updates (2500 here, vs. ∼14000 for Goyal et al 2007)

---

> ### Author Response · Authors · 2017-12-21
> **Response to review**
>
> We thank the reviewer for their positive assessment of our work.
> To respond to the comments raised:
>
> The wall clock time is primarily determined by the hardware researchers have at their disposal; not the quality of the research/engineering they have done. In the paper we choose to focus on the number of parameter updates, because we believe this is the simplest and most meaningful scientific measure of the speed of training. Assuming one can achieve perfect parallelism, the number of parameter updates and the wall clock time are identical. However we can confirm here that, using the increasing batch size trick, we were able to train ResNet-50 to 76.1% validation accuracy on ImageNet in 29 minutes. With a constant batch size, we achieve comparable accuracy in 44 minutes (replicating the set-up of Goyal et al.). This significantly under-estimates the gains available, as we only increased the batch size to 16k in these experiments, not 64k as in the paper.
>
> One of the goals of large batch training is to remove the need for asynchronous SGD, which tends to slightly reduce test set accuracies. Since we are now able to scale the batch size to several thousand training examples and train accurate ImageNet models in under an hour with synchronous SGD, the incentive to use asynchronous training is much reduced. Intuitively, asynchronous SGD behaves somewhat like an increased momentum coefficient, averaging the gradient over recent parameter values.
>
> We chose to focus on clarity, rather than including many equivalent experiments under different architectures, however we have checked that our claims are also valid for a DNN on MNIST and ResNet-50 on ImageNet. This is also why we do not present an exhaustive range of learning rate scales in section 5.2; we wanted to keep the figures clean and easy to interpret. It's worth noting that our observations also match theoretical predictions. We will update the figure captions to clarify which model/dataset they refer to.

---

> > ### Comment · AnonReviewer2 · 2018-01-12
> > **Be careful drawing too many sweeping conclusions**
> >
> > I would be very cautious about claims that because we can succeed on ImageNet in an hour, the incentive to care about distributed/asyncronous SGD is "much reduced". The world of machine learning is so much bigger than deep convolutional networks for images, and the idea that we can solve all big data problems (or most) via single-machine SGD makes little sense to me.
> >
> > I would definitely encourage the authors to consider some models/datasets that are not related to images at all. This would be a much more sincere test about generalization, rather than just trying out MNIST.
> >
> > RE many vs few learning rate values: I am in agreement that main paper figures should focus on clarity. But I do think there needs to be some careful experiments, documented in a supplement, that are more exhaustive. I also especially suggest this with regard to my comment about reporting performance across many random initializations (with several different initialization strategies). From the current results, a careful reader might not know whether to call the conclusions "reproducible" or just "lucky".
> >
> > Thanks for providing the wallclock speedup measurements. Of course wallclock time is dependent on hardware, but I do like to see comparisons of different methods on the same hardware for assessing practical utility. I would still suggest including these in the paper (with appropriate caveats and clarification of specific hardware used). I would not find "number of parameter updates" alone to be persuasive evidence. What if the cost of all those parameter updates is dwarfed by other factors? I would also drop the argument that "assumes perfect parallelism", because I've never met a real system that was close to near-perfect parallelism.

---

> > > ### Author Response · Authors · 2018-01-22
> > > **Clarifications**
> > >
> > > We apologize for any confusion. We do not claim that most big data problems can be solved using single machine SGD; our experiments use distributed (but synchronous) SGD. We will edit the text to clarify that the incentive to use asynchronous training is reduced when the synchronous batch size can be scaled to a substantial fraction of the training set size (as observed in our paper). Although we expect this to be commonly the case, we acknowledge that it is not guaranteed to occur for all models/datasets.
> > >
> > > We will add our additional results reporting wall-clock times on ResNet-50 (likely in an appendix). We will also update the section 5.2 to provide median of 5 runs for each case shown, and add experiments at a wider range of learning rate values to the appendix.
> > >
> > > We state that parameter updates provide a measure of the training speed if one assumes near-perfect parallelism, since this clarifies how readers should interpret our results depending on their own hardware/scaling efficiency. We note that a number of recent works have achieved extremely good scaling. Eg Goyal et al. [1] report large batch scaling efficiency of 90% while Akiba et al. [2] report scaling efficiency of 80%; both compared to a single node of 8 GPUs. On our own hardware we can increase the batch size from 256 to 16k with >95% scaling efficiency.
> > >
> > > [1] "Accurate, Large Minibatch SGD: Training ImageNet in 1 Hour", Goyal et al., 2017, arXiv:1706.02677
> > > [2] "Extremely Large Minibatch SGD: Training ResNet-50 on ImageNet in 15 Minutes", Akiba et al., 2017, arXiv:1711.04325

---

### Public Comment · (anonymous) · 2017-11-13
**A Suggestion for Evaluation Setting**

The authors claimed that one can achieve equivalent test accuracies by increasing the batch size proportionally instead of decaying the learning rate. They also claimed that one benefit of increasing batch size is it has fewer parameter updates. However, to make the latter claim more convincing, it is strongly suggested adding a comparison with a fixed-size large batch method (say, much larger than the initial batch size of the "Increasing batch size" method) in the evaluation setting, since large batch method may have even fewer updates than the "Increasing batch size" method. If the large batch method cannot reach same test accuracies after the same number of training epochs despite fewer updates, then the claim that "Increasing batch size" method can achieve equivalent test accuracies with fewer updates than fixed batch size method can be solidly confirmed.

I am quite interested in work on changing batch sizes and found one paper introducing ways to dynamically adapt batch size as learning proceeds, called "On Batch Adaptive Training for Deep Learning: Lower Loss and Larger Step Size" (also submitted to ICLR 2018). They've done similar work but in a self-adaptive way. Specifically, it proposed a method to dynamically select the batch size for each update so that it may achieve lower training loss after scanning the same amount of training data. However, its batch adaptive method requires more computation costs to decide a proper batch size. Check it out if you are interested.

---

> ### Author Response · Authors · 2017-11-23
> **Extra results in appendix after review**
>
> Thank you for your interest in our work!
>
> We would like to emphasize that the method of increasing the batch size during training instead of decaying the learning rate is not an alternative to large batch training, it is complimentary. Large batch training is achieved by increasing the initial learning rate and linearly scaling the batch size, thus holding the SGD noise scale constant. Meanwhile we propose increasing the batch size during training at constant learning rate, in order to maintain the same noise scale progression obtained by a decaying learning rate.
>
> We showed in figure 5 that we could simultaneously increase the initial learning rate and batch size by a factor of 5, and also replace a decaying learning rate with an increasing batch size schedule. This did not cause any reduction in test performance, achieving final test accuracy of 94.5%. In response to your question, we attempted to instead increase the initial learning rate and batch size by a factor of 25 with a constant batch size and decaying learning rate schedule. The test set accuracy drops to 93.2%. We will add these results to the appendix after the review process.
>
> Best wishes,

---

### Author Response · Authors · 2018-01-05
**Updated manuscript**

We have uploaded an updated manuscript, responding to the comments of the referees. We were delighted that all three reviewers recommended the paper be accepted. As well as fixing some minor typos, the main changes are:

1) We have edited the final paragraph of section 4, to clarify that the performance losses with large batches/momentum coefficients are not resolved by using initialization bias correction suggested by reviewer 3. When the momentum coefficient is too large, it takes many epochs for the accumulation to forget old gradients, and this prevents SGD from responding to changes in the loss landscape.

2) We clarify in section 5.3 that we chose not to include wall-clock times, since these are not comparable across different hardware/software frameworks. As we stated in response to reviewer 2, we have confirmed that increasing batch sizes can be used to reduce wall-clock time.

3) We include a brief discussion of asynchronous SGD in the related work section.

---

### Decision · Program_Chairs · 2018-01-29
**ICLR 2018 Conference Acceptance Decision**

**Decision:**

Accept (Poster)

**Comment:**

Pros:
+ Nice demonstration of the equivalence between scaling the learning rate and increasing the batch size in SGD optimization.

Cons:
- While reporting convergence as a function of number of parameter updates is consistent, the paper would be more compelling if wall-clock times were given in some cases, as that will help to illustrate the utility of the approach.
- The paper would be stronger if additional experimental results, which the authors appear to have at hand (based on their comments in the discussion) were included as supplemental material.
- The results are not all that surprising in light of other recent papers on the subject.